# A Sequence of Phase Transformations and Phases in NiCoFeCrGa High Entropy Alloy

**DOI:** 10.3390/ma14051076

**Published:** 2021-02-25

**Authors:** Ádám Vida, János Lábár, Zoltán Dankházi, Zsolt Maksa, Dávid Molnár, Lajos K. Varga, Szilvia Kalácska, Márk Windisch, Gabriella Huhn, Nguyen Q. Chinh

**Affiliations:** 1Department of Industrial Materials Technology, Bay Zoltán Nonprofit Ltd. for Applied Research, Kondorfa u.1., H-1116 Budapest, Hungary; Adam.Vida@bayzoltan.hu (Á.V.); Mark.Windisch@bayzoltan.hu (M.W.); 2Department of Materials Physics, Eötvös Loránd University, P.O.B. 32, H-1518 Budapest, Hungary; labar.janos@ek-cer.hu (J.L.); zoltan.dankhazi@ttk.elte.hu (Z.D.); zs.maksa@gmail.com (Z.M.); huhnandrasne@gmail.com (G.H.); 3Thin Films Physics Laboratory, Centre of Energy Research, Konkoly-Thege u. 29-33, H-1121 Budapest, Hungary; 4Materials Science Group, Dalarna University, SE-791-88 Falun, Sweden; mdi@du.se; 5Applied Materials Physics, Department of Materials Science and Engineering, Royal Institute of Technology, SE-100-44 Stockholm, Sweden; 6Institute of Solid State Physics and Optics, Wigner Research Center for Physics, H-1121 Budapest, Konkoly-Thege u. 29-33, Hungary; varga@szfki.hu; 7Empa, Swiss Federal Laboratories for Materials Science and Technology, Laboratory of Mechanics of Materials and Nanostructures, Feuerwerkerstrasse 39, 3602 Thun, Switzerland; szilvia.kalacska@empa.ch

**Keywords:** NiCoFeCrGa high entropy alloys, phase transformations, multiphase, spinodal decomposition, magnetic behavior

## Abstract

The present investigation is directed to phase transitions in the equimolar NiCoFeCrGa high entropy alloy, which is a mixture of face-centered cubic (FCC) and body-centered cubic (BCC) crystalline phases. The microstructure of the samples was investigated by using scanning electron microscopy (SEM), time-of-flight secondary ion mass spectroscopy (TOF-SIMS), transmission electron microscopy-based energy-dispersive spectroscopy (EDS) and electron energy loss spectroscopy (EELS), as well as X-ray diffraction (XRD) measurements. Based on the phases observed in different temperature ranges, a sequence of the phase transitions can be established, showing that in a realistic process, when freely cooling the sample with the furnace from high to room temperature, a microstructure having spinodal-like decomposition can also be expected. The elemental mapping and magnetic behaviors of this decomposed structure are also studied.

## 1. Introduction

Being multicomponent alloys, the microstructure of the high entropy alloys (HEAs)—in principle—could be highly variable. Surprisingly, however, most of the previous investigations have shown that the structure of the HEAs is quite simple: the mixture of face-centered cubic (FCC) and body-centered cubic (BCC) or hexagonal close-packed (HCP) lattices. There are no, or just a very few, intermetallic phases in the microstructure of HEAs. Based on the high number of elements, one might think that there should exist more likely and unlikely atomic groups due to the enthalpies of forming, but the system remained in a solid solution state, at least it seemed so for a long time [1,2,3]. In addition, it should be mentioned that the cooling rate may have a significant effect on the microstructure. It has been shown in the case of the NiCoFeCrGa high entropy alloy [4,5] that the cooling rate influences both the composition of the two (FCC and BCC) phase-components and the ratio of their volume fractions.

Considering the high number of possible alloying combinations, the high entropy stabilization is believed to be the key feature of the HEAs. Accordingly, many papers were dealing with the question of entropy from the beginning [6,7,8] and also in the later years [9,10,11,12,13,14,15,16,17]. The later publications were mainly focusing on the phase stability of single-phase alloys; therefore, the investigations were carried out after applying the right heat treatments. From the discussions of these papers, some conclusions can be drawn: (i) the as-cast state should be considered as metastable, and (ii) only very few alloys show resistance against heat treatments and remain single-phase.

The four-component, equimolar NiCoFeCr, often called the first-and-base HEA, has been the topic of recent theoretical and experimental works [3,18,19,20,21,22,23]. Furthermore, the systematic investigation of the effect of the sp. *element* additions is also well studied in the literature [24,25,26,27].

Without adding any doping elements, the structure of NiCoFeCr base alloy is a single-phase FCC in a paramagnetic state at room temperature. It has been shown that with the addition of Ga, the single-phase base system turned into a duplex in the equimolar NiCoFeCrGa high entropy alloy [26]. The role of phase mixing in this alloy was then studied and analyzed by tracking the mechanical and structural properties of the alloy obtained after different heat-treatments [4]. Together with the microstructural investigation, it was also observed that the NiCoFeCrGa system shows anomalous thermal expansion, which can be explained by the phase transition from ferromagnetic BCC to paramagnetic FCC in the microstructure of the alloy [27].

Continuing previous works, the goal of the present study was to follow the evolution of phase structures in the NiCoFeCrGa HEA during targeted heat treatments. In this work, we report for the first time on a sequence of phase transitions and on the spinodal-like decomposition in this high entropy alloy.

## 2. Materials and Methods

### 2.1. Processing of the Samples

The investigated alloys were prepared from pure metals (99.99%) with high precision in Wigner Research Center for Physics, Budapest Hungary. Induction melting in a cold copper mold with an inert atmosphere was used to prepare the specimens. Every ingot was at least five times re-melted to help the mixing. The typical mass of the specimen was around 25 g.

The specimen was first ground to obtain a flat surface, then mechanically polished by 9 µm, 6 µm and 1 µm particle size diamond suspensions. The last step of the polishing was done by colloidal silica (Struers OP-S and OP-U). The heat treatments were performed in a tube furnace under Ar gas (99.95% purity) protection. Separate samples were heat-treated for 1 h at different temperatures in the range between 560 and 1273 K and then quenched into the water at room temperature (WQ). The heating rate was set to 10 K/min. An initial sample was treated at 1150 K for 1 h and then slowly cooled (SC) with the furnace. Here we note that the cooling speed of the as-cast (AC) sample during the casting was around 50 K/s. Furthermore, the cooling speed of WQ and SC samples was around 10^3^ and 10^−1^ K/s, respectively [4].

### 2.2. Microstructure Characterization Techniques

The microstructural investigation and chemical analysis were carried out with scanning electron microscopes (SEM) in multifunctional FEI Quanta 3D equipment (Thermo Fischer Scientific, Waltham, MA, USA) and a Zeiss ULTRA 55 (Zeiss, Jena, Germany). Electron backscattered diffraction (EBSD) was also carried out with Quanta 3D.

For the chemical surface analysis, a Tescan Lyra3 FEG-SEM (TESCAN, Brno, Czech Republic) was used, equipped with a time-of-flight secondary ion mass spectroscopy (TOF-SIMS, TOFWERK AG. Thun, Switzerland). This focused ion beam (FIB) based method [28,29,30] uses the Ga^+^ impacting ions to sputter the sample locally while collecting and selecting the sputtered ions based on their mass. TOF-SIMS has the advantage of excellent lateral and depth resolution, as the sputtered ions originate from the few top atomic layers [31,32]. However, the sputtering speed depends on the chemical and crystallographic composition of the sample; therefore, surface roughness analysis after TOF-SIMS measurement can give qualitative information about the sputtering speeds of the analyzed structures [33]. TOF-SIMS measurements were concluded using an ion beam of 30 kV, 40–160 pA combined with a gas injection system (GIS) operating with fluorine gas. The fluorine gas was applied to increase ionization probability to enhance secondary ion signals during TOF-SIMS analysis [34]. Secondary electron images were recorded before and after FIB sputtering with a 20 kV electron beam at high contrast conditions.

The 300 keV transmission electron microscopy (TEM) was performed on a JEOL 3010 (JEOL, Tokyo, Japan), equipped with an electron energy loss spectrometer (EELS). During the Cr mapping, a Gatan GIF Tridiem energy-filtered mode was applied. This mode is called energy-filtered transmission electron microscopy (EFTEM).

Thermal analysis was performed with a differential thermal analyzer (DTA, SETARAM SetSys, KEP Technologies, Mougins, France) from room temperature to 1573 K (1300 °C) at a heating rate of 10 K/min in Ar atmosphere. The sample holder was alumina, which was first heated empty to determine the baseline.

The local composition was further determined by using auger electron spectroscopy (ES, Physical Electronics, Chanhassen, Minnesota) technique on a PHI 700 xi. Data were collected using 20 kV accelerating voltage and 10 nA beam current.

The phase compositions were investigated by X-ray diffraction (XRD) using a Philips Xpert θ–2θ powder diffractometer (Phillips, Amsterdam, Netherlands) with CuKα radiation (wavelength: λ = 0.15418 nm). The pattern was collected using a tube voltage of 40 kV and tube current of 30 mA. The lattice parameter was determined by extrapolating the lattice parameters obtained from the different reflections to the diffraction angle of 2θ = 180° using the Nelson–Riley method [34].

### 2.3. Magnetic Measurements

Magnetic force microscopy (MFM) was used to investigate the local magnetic properties of the samples. The phase characteristic of the cantilever’s vibration was detected during the MFM measurements carried out by using a Solver Pro NT-MDT atomic force microscope (AFM, NT-NDT, Moscow, Russia). The AFM topography and MFM imaging were performed simultaneously in the same area of the sample by using the two-pass technique. The tip was lifted above the surface of the sample by a distance of about 60 nm during the MFM measurement. For the dynamic detection mode, the cantilever was vibrating with its resonant frequency (85 kHz), and the phase portion may be modulated by the magnetic forces emerging from the sample. The applied magnetic tips were microfabricated Si cantilevers with a pyramidal tip coated with magnetic Co-Cr thin film having 40 nm thickness.

## 3. Results and Discussion

### 3.1. General Observations

Figure 1 shows the differential thermal analysis (DTA) thermogram of the initial AC sample. Considering the main processes, except for the multistep melting of the complex system (marked as region D), visible phase-transitions can be expected in the two exothermic regions C and B at 1000–1200 and 900–1000 K temperature ranges, respectively. It has been shown before that the change of heat-flow observed in region A is related to the surrounding of the Curie point of this alloy [4]. The average melting temperature, *T_m_*—estimated by the concentration weighted average rule—of this equimolar NiCoFeCrGa sample was found to be 1547 K, which is in good accordance with the experimentally measured value.

As mentioned above, in order to study the possible microstructures and the evolution of the phases, the initial (AC) samples were heat-treated for 1 h at different temperatures and then quenched into the water at room temperature (WQ). An initial sample was treated at 1150 K and then slowly cooled with the furnace (SC).

Typical microstructures of the initial (as-cast, AC) and the WQ samples can be seen in the SEM images of Figure 2.

The images were taken by using the backscattered electrons, making the contrast by different atomic compositions of phases, showing the typical dendrite features characterizing the microstructures of the samples heat-treated at different temperatures and water-quenched. Similar to previously reported results [4,5], the experimental results reveal a global microstructure with FCC-phase dendrite and BCC-phase inter-dendrite, as marked in the images. Typical compositions of these phases can be found in the previous works [4,5]. The general picture is somehow changed in the case of the sample treated at 1150 K (see Figure 2e), where BCC-phase needle-shaped particles were formed inside FCC regions. The formation of these particles is certainly corresponding to the exothermic peak C in the thermogram of Figure 1.

More detailed SEM investigations have revealed that in the case of the WQ sample aged at 923 K, the BCC-phase regions of the sample are decomposed into small, cuboidal-shaped particles having a size between 50 and 150 nm, as presented in Figure 3. In the figure, both the lower magnification image (already shown in Figure 2d) and the higher magnification, one on the WQ sample aged at 923 K, can be seen. This spinodal-like decomposition is certainly corresponding to the other, lower exothermic peak B in the curve of Figure 1.

Furthermore, experimental results have also shown that in the case of SC sample heat-treated at 1150 K and then slowly cooled with the furnace to room temperature, the resulting microstructure—also presented in Figure 3 for comparison—shows similar features to that observed in the two WQ samples aged at 1150 K and at 923 K, mentioned above. Both the global view showing the mixture of FCC and BCC regions with the BCC-phase needle-shaped particles and the local spinodal-like decomposition of the BCC regions into small particles can be observed in this sample (see Figure 3c,d). Since the obtained structure has not been observed in the literature so far, further TEM and SEM investigations were performed to know as many details as possible about the particles (see later). Here we note that the results of EFTEM investigations indicate an enhanced amount of Cr inside the cuboidal-shaped particles (see in Figure 3e).

Figure 4 shows the XRD profiles taken on some investigated samples. The results of the XRD measurements are in good agreement with that of SEM investigation, indicating that the general structure of all heat-treated samples remains unchanged, always containing a mixture of FCC and BCC lattices. Without detailed analysis, it can also be seen that the XRD profiles reflect a significant difference between the microstructures of the WQ and SC samples aged at 1150 K but show many similarities between the structure of the SC sample and that of the WQ sample aged at 923 K.

### 3.2. A Possible Sequence of Phase Transformations

Due to their high entropy, the microstructure of HEAs would be very complicated. However, in a realistic process, when the sample is allowed to cool together with the furnace, the mentioned microstructure formed from a phase transition at about 1150 K, in region C, followed by decomposition at about 900 K, in region B in Figure 1 remains stable until the sample has cooled to room temperature. Accordingly, a schematic sequence of the evolution of the phase structure in NiCoFeCrGa high entropy alloy is suggested in Figure 5.

Distinctive structural and chemical changes are visible in Figure 5. In the high-temperature state, the entropic term (*T*Δ*S*) of Gibbs-equation (Δ*G* = Δ*H* − *T*Δ*S*) is high enough due to high-temperature (in the region D of the thermogram shown in Figure 1). This phenomenon plays a key role in stabilizing the near 50–50% phase mixture of the two solid solution phases [4]. When the temperature decreases, the entropic term also lowers, so the enthalpy term (Δ*H*) becomes more significant in the balance due to the chemical/lattice energies. This first leads to a crystal structure change, where a significant proportion of the FCC structure undergoes a transformation to needle-like BCC structures. It is worth mentioning that the chemical composition of the BCC-phase needle-shaped particles located inside the FCC regions is similar to that of the original BCC regions formed from the high-temperature state (in the region C in Figure 1) [4]. Furthermore, with lowering the temperature, a spinodal-like decomposition takes place with the formation of small cuboidal particles inside BCC regions (in the region B in Figure 1). Further investigations are needed to study in detail the whole phase-transition process. In the present case, the experimental results suggest that continuing the cooling, no more change is visible in the system; thus, the state developed around 900 K can be attributed to that room temperature one.

### 3.3. Microstructure of the Spinodally Formed Precipitates

Considering the microstructures of the investigated NiCoFeCr-based HEA alloys, the mixture of the relatively large FCC and BCC regions, as well as the presence of the needle-like BCC phase precipitates, are well-established. However, as mentioned, the spinodal-like decomposition of the BCC phase together with the formation of the small cuboidal particles is a new experimental observation. Hence, far, we have no knowledge of such phase-transition. This is a strong motivation for us to see in more detail both the microstructure and magnetic behavior of these small particles.

Figure 6 shows a bright-field TEM image on the BCC region containing cuboidal precipitates of the SC sample (Figure 6a) and the corresponding selected area electron diffraction (SAED) pattern (Figure 6b). The indexed SAED pattern corresponds to the BCC phase from the zone-axis <331>. The cuboidal precipitates do not have a different structure or orientation, as they did not cause extra spots in the diffraction pattern (although their volume fraction is high, as seen from the BF image). In addition to the bcc spot, only thermal diffuse scattering (TDS) streaks can be seen in the SAED pattern.

The mentioned similarity in structure and orientation of the cuboidal precipitates to the parent matrix also has been confirmed by the results of EBSD investigation, shown in Figure 7, where inverse pole figure map FCC and BCC phases can be seen.

A typical microstructure—containing extended regions of FCC and BCC phases, as well as needle-like and cuboidal particles—of the microstructure of the SC sample heat-treated at 1150 K for 1 h can be seen in this figure. The BCC nature of both needle- and cuboidal precipitates is well perceptible in the inverse pole figure (IPF) presentation. Furthermore, no phase boundary is observable between the cubes and the parent matrix, indicating unambiguously that these cuboidal particles are created only by segregation and/or migration of chemical elements with the possible spinodal mechanism.

The obtained results would suggest a significant difference between the composition of the cuboidal particles and the surrounding matrix.

Elemental mappings taken by TOF-SIMS on the characteristic spinodal-like decomposition are shown in Figure 8. Significantly increased ^52^Cr ion content is detected in the small cuboid precipitates. This is in good agreement with the result of the EFTEM measurement shown in Figure 3e. Along with the local increment of Cr, the decrease of the other components can be observed when looking at the cuboids. At the same time, the BCC matrix has a higher concentration of all other ^56^Fe, ^58^Ni, ^59^Co and ^69^Ga ions, while ^52^Cr ion depletion is evident in the same area. Interestingly, differences between the FCC and BCC matrix can also be seen in Figure 8. The concentration of ^56^Fe ions seems to be less affected by the phase boundary than ^58^Ni, ^59^Co and ^69^Ga ions, where a decrease in the FCC phase appears.

Considering the quantitative comparison, Figure 9 shows the typical ES results of elemental analysis for both the cubes and the matrix. The chemical composition of two cubes and of two local surrounding BCC matrix-areas is shown in Figure 9a. It can be seen that besides the decrease of Ni, there is a significant—up to 40%—increase of Cr concentration in the cubes. For completeness, the composition of the FCC phase is also shown in Figure 9b. Here we note that as the ES measurements are carried out to determine the chemical composition near the surface, the obtained concentration of Ga is systematically lower than the nominal value. This is presumably due to the fact that Ga is a low melting point metal (its melting point is only 30 °C), and due to its partial surface evaporation—during measurement—the detectable amount will be less than the expected one. In any case, the increased amount of Cr within the cubes is a proven fact.

Figure 10 presents the AFM and MFM images taken on the same slowly cooled sample, showing all four main features—FCC, BCC regions, BCC-phase needle-shaped precipitates and cuboidal particles—of the microstructure of this sample. Results of the magnetic measurements reveal that despite the magnetic behavior of BCC-phase in general, the cuboidal particles behave like non-magnetic islets due to the high Cr content inside the BCC-phase regions. Although further studies are needed, it can be seen that the formation of small particles can determine not only the mechanical but also the magnetic properties of the material, broadening the potential applications of high entropy alloys.

As mentioned before, due to their high entropy, the microstructure of HEAs would be very complicated. Although at this moment, no thermodynamic calculation is available to rationalize the suggested sequence of phase transformations, the obtained results will hopefully provide important experimental experiences to better understand several behaviors of HEAs, such as the mechanical, the corrosive properties and even the diffusion behavior of the individual elements.

## 4. Conclusions

Phase transitions in the NiCoFeCrGa HEA were studied by completing different heat-treatments and investigating the microstructure using several methods. According to the experimental results, a sequence of the phase transitions can be established, describing the main steps of a realistic process when freely cooling the sample with the furnace from high to room temperature. At high temperatures higher than 1200 K, both FCC and BCC solid solution phases are presented in equal fractions in the material. Lowering the temperature to around 1100 K, BCC-phase needle-shaped precipitates are formed in FCC regions. Going down to around 900 K, a spinodal-like decomposition takes place with the formation of Cr-rich, small-size cuboidal particles, which may determine both the mechanical and magnetic behaviors of the material, broadening the potential applications of high entropy alloys.

## Figures and Tables

**Figure 1 materials-14-01076-f001:**
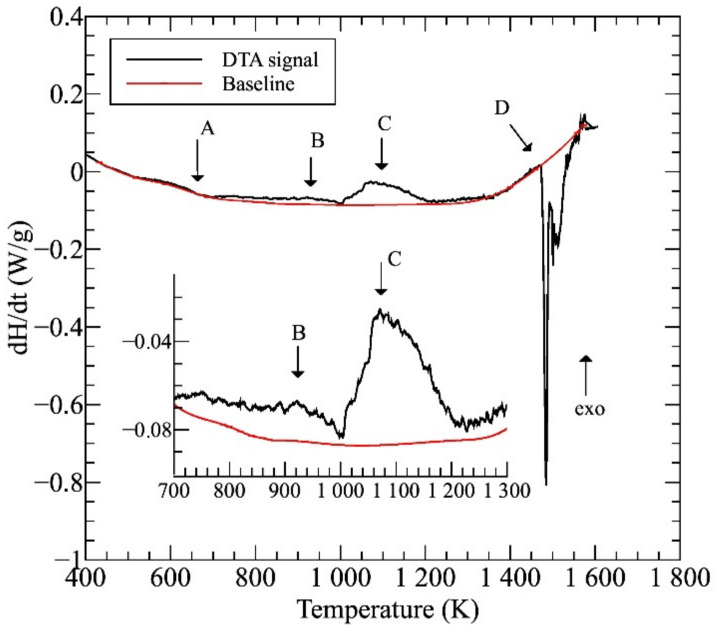
The differential thermal analysis (DTA) thermogram taken on the initial (AC) sample.

**Figure 2 materials-14-01076-f002:**
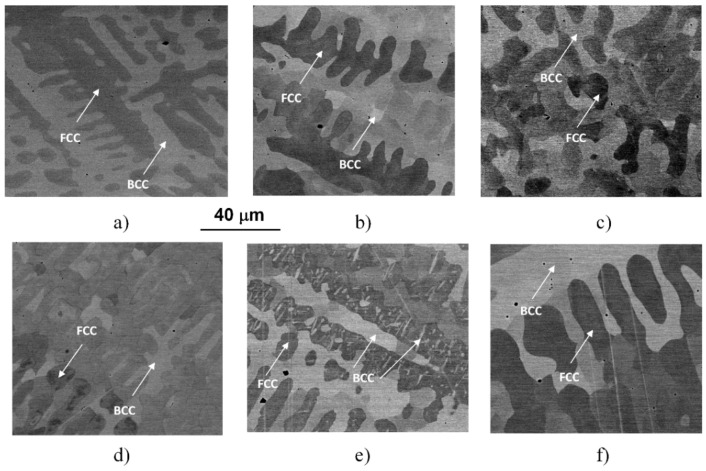
SEM images taken on the initial (AC) state (**a**) and on the samples heat-treated for 1 h at 560 (**b**), 700 (**c**), 923 (**d**), 1150 (**e**) and 1273 K (**f**) and quick-cooled by water-quenching (WQ).

**Figure 3 materials-14-01076-f003:**
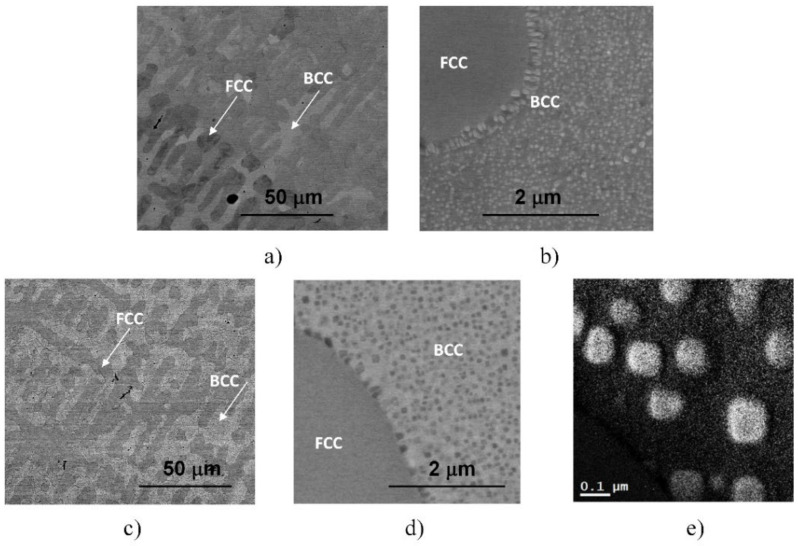
SEM images taken on the samples heat-treated for 1 h at 923 K, quick-cooled by water quenching (**a**,**b**) and at 1150 K, but slowly cooled with the furnace (**c**,**d**). The microstructure of the later sample was also investigated by EFTEM, indicating an enhanced amount of Cr inside the cuboidal-shaped particles in the Cr-map (**e**).

**Figure 4 materials-14-01076-f004:**
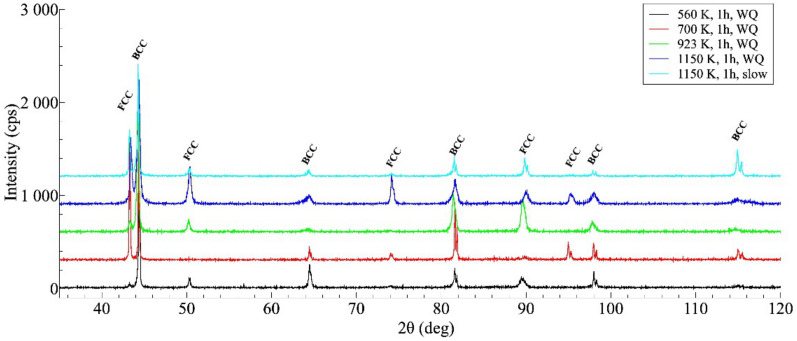
XRD profiles taken on the samples after different heat-treatments, showing the microstructure having face-centered cubic (FCC)/ body-centered cubic (BCC) duplex after each heat treatment (the curves are shifted vertically by about 300 units from each other for a better view).

**Figure 5 materials-14-01076-f005:**
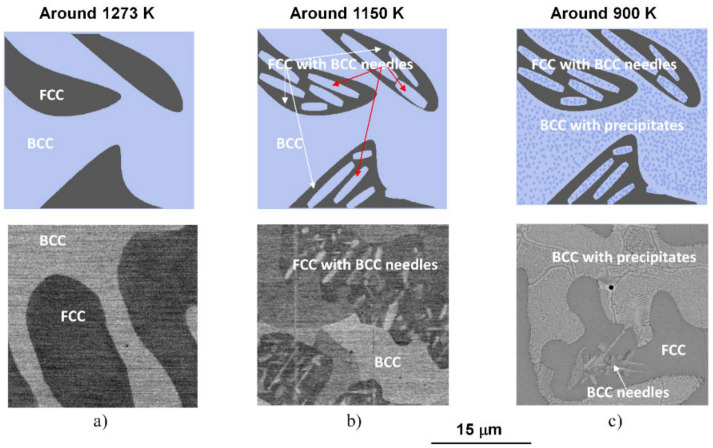
Schematic sequence of the evolution of the phase structure during slow cooling (top row) and the corresponding SEM micrographs (bottom row) taken on the samples heat-treated under conditions of (**a**) 1273 K for 1 h and WQ, (**b**) 1150 K for 1 h and WQ and (**c**) 1150 K for 1 h and slowly cooling with the furnace.

**Figure 6 materials-14-01076-f006:**
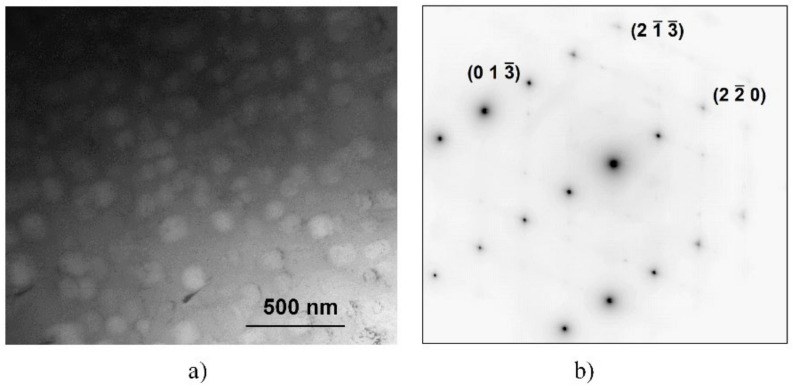
Microstructure of the BCC region of the SC sample heat-treated at 1150 K for 1 h and slowly cooled, shown by bright-field TEM (**a**) and corresponding selected area electron diffraction (SAED)-pattern (**b**) from the zone-axis of orientation <331>.

**Figure 7 materials-14-01076-f007:**
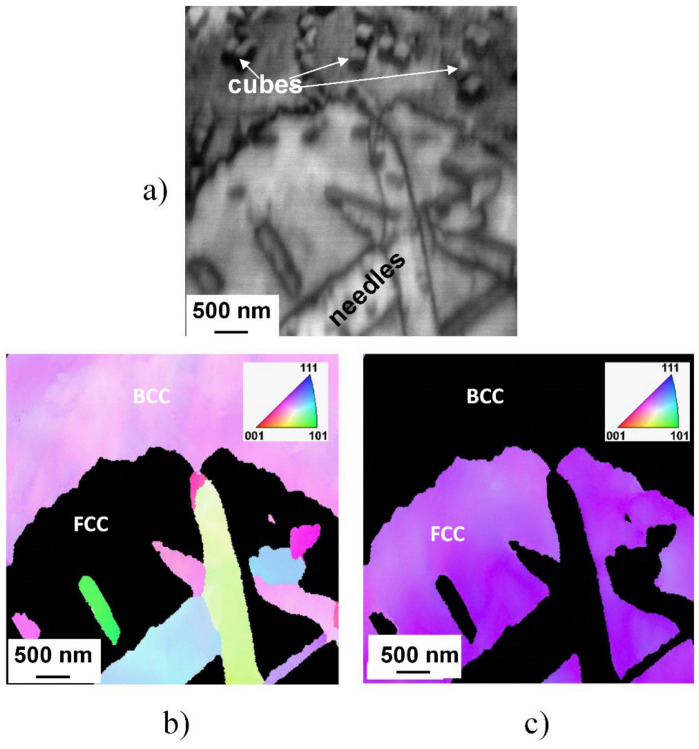
Typical microstructure of the slowly cooled (SC) sample heat-treated at 1150 K for 1 h and slowly cooled, demonstrated by image quality plot build from electron backscattered diffraction (EBSD) data (**a**) and the corresponding inverse pole figure map of FCC (**b**) and of BCC (**c**) phases.

**Figure 8 materials-14-01076-f008:**
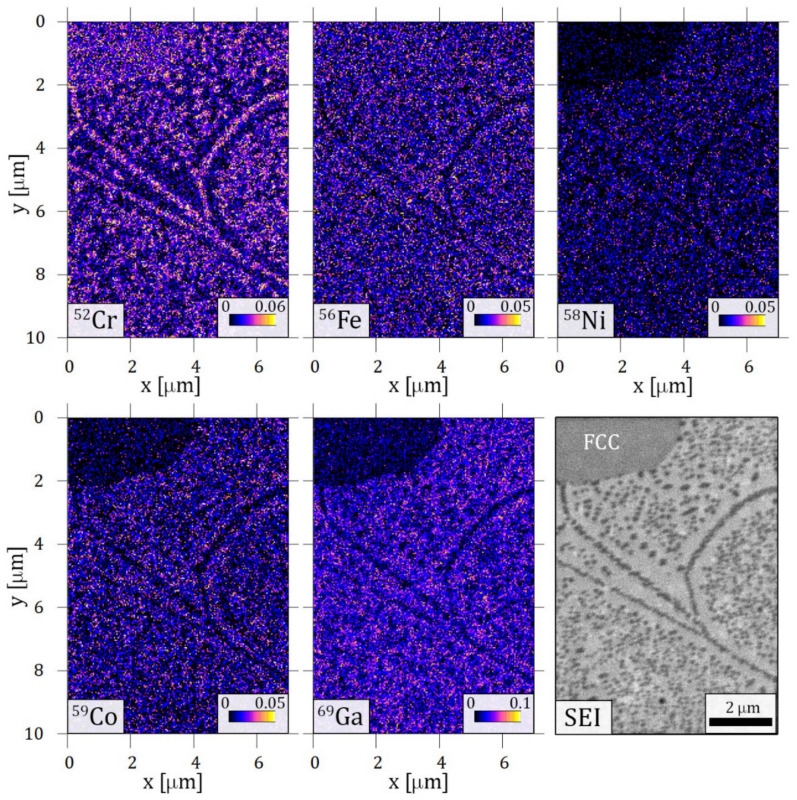
Elemental maps built from time-of-flight secondary ion mass spectroscopy (TOF-SIMS) results of the sample heat-treated at 1150 K for 1 h and slowly cooled. The area contains a phase boundary of BCC with cuboid structure and FCC phase (marked on the secondary electron image, SEI).

**Figure 9 materials-14-01076-f009:**
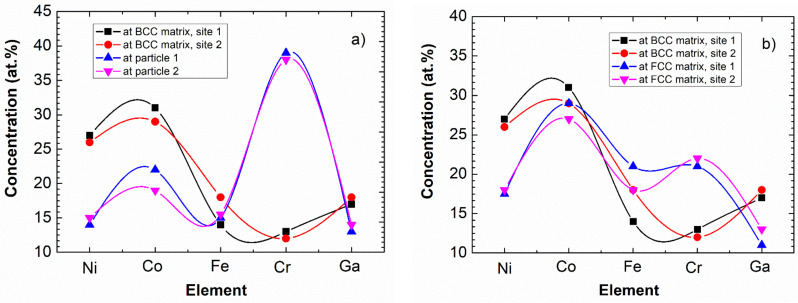
Composition of cuboidal particles and the surrounding BCC matrix (**a**), as well as that of BCC matrix and FCC phase (**b**). The data determined by ES having a relative error of about 5%. Note that the connecting lines are only indicating the elements belonging to the same measured points for easier clarity.

**Figure 10 materials-14-01076-f010:**
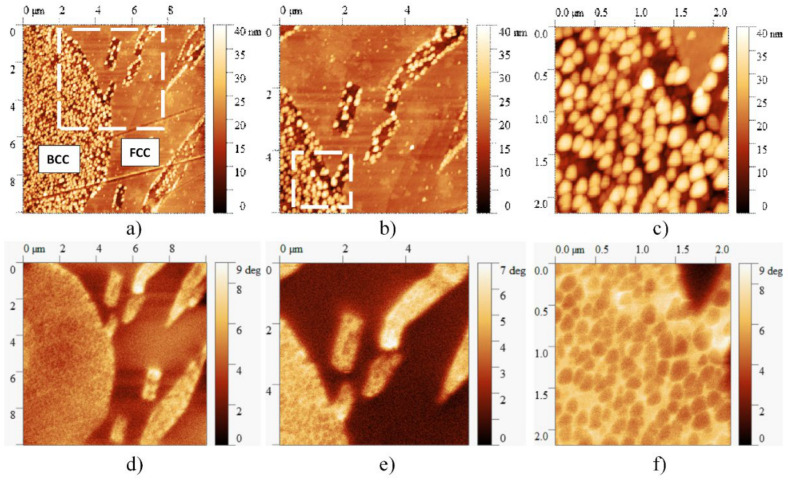
AFM (**a**–**c**) and corresponding MFM images (**d**–**f**) taken at different magnifications on the microstructure of the sample heat-treated at 1150 K for 1 h and slowly cooled. On the MFM image, the brighter color represents the higher magnetic property. Despite the magnetic behavior of the BCC phase, the cuboidal particles behave like non-magnetic regions due to the high Cr content.

## Data Availability

The data that support the findings of this study are all own results of the authors, not available anywhere.

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
