# Peer review of "A Sequence of Phase Transformations and Phases in NiCoFeCrGa High Entropy Alloy"

_materials, 2021, doi:10.3390/ma14051076_

Round 1
Reviewer 1 Report
The authors of this study investigate the phase transformations of a five-component alloy. A few different heat treatments coupled with simple electron microscopy and magnetic measurements are used to elucidate the decomposition processes occurring at various temperatures. The study in its present form does not perform a rigorous assessment of their experimental results. Further, the authors do not make any efforts to provide insights into the relationships between the structure and the properties of this alloy. The study in its current form contains several scientific deficiencies and for this reason I cannot recommend publication.
- Phase identification throughout the manuscript seems to be performed based on only contrast arising from EBSD micrographs. It is well-known that such contrast cannot provide a rigorous identification of the crystal structure. The XRD patterns of fig. 4 provide a general idea of the phases that may be present but do not identify which of the phases would be enriched in the heavier elements (thus providing contrast). Could the authors provide rigorous diffraction data to justify their identification?
- The authors assert that the "spinodal-like" decomposition is associated with the "lower exothermic peak B in the curve of Fig. 1". The alloys used to present figures 2 and 3 have been subjected to aging heat treatments and subsequent cooling treatments while the alloy used to generate the DTA curve of figure 1 was subject to a completely different processing protocol. Given the differences in processing conditions could the authors justify associating the cuboidal precipitates of fig. 1 with the phase transition they observe?
- The authors make no effort to identify the precipitates that are observed in figures 3b,d, and e. Can they demonstrate how and why they think these precipitates have the same crystal structure as the matrix phase? Could they also provide some insights into what this phase might be? The authors claim that they "investigated by TEM EELS" the cuboidal precipitates in fig. 3e and it showed "very significant presence of Cr". Could they show these results and quantify what they mean by "significant"?
- Could the authors provide any thermodynamic calculations to rationalize the phase transformation sequence that they observe?
Reviewer 2 Report
This work describes using annealing microstructures at different temperatures to derive phase transformation sequence during cooling. The bi-modal structured BCC phase seems interesting. But I think quite a few points as listed below need to be clarified or supplemented and the language needs to be polished before this manuscript is qualified for the journal Materials.
- Please specify under what condition the sequence of phase transformation is investigated. What does cooling specifically refer to?
- Please provide detailed information about the material and heat-treating processes in Section 2.
- Please provide detailed information about DTA measurement, including baseline measurement.
- The contrast for FCC and BCC phases in Figure 2, Figure 3 and Figure 4 are different. Please specify what type of signals were detected during SEM, and label phases in these figures. Figure 3b and Figure 3d display opposite contrast, making them very confusing.
- Compositions of FCC phase and bi-modal BCC phases are needed to better distinguish the phases in SEM images and also to help understand ‘spinodal-type decomposition’.
- Since the mechanism of the formation of cuboidal BCC phase is not determined in this work, I would refrain from mentioning ‘spinodal-like decomposition’ in the abstract. Instead, one could mention cuboidal BCC structure.
- Please provide the micrograph showing the microstructures of the as-cast sample.
- Please rationalize using annealed structures to derive cooling microstructures. It is more rational to cool from high temperature (solutioned) to a specific lower temperature T and then quench to retain the microstructures at T.
Reviewer 3 Report
This is an interesting article, describing the phase transformation process of a NiCoFeCrGa high entropy alloy. Phase transformation during heat treatment processes and under loading is very important for multi-phase high entropy alloys, since it affects the mechancial properties and other properties significantly. However, more detailed characterizations should be added for the paper, since no detailed transformation process and mechanisms can be found in present results. TEM, particularly HRTEM, might help a lot to understand the transformation mechanisms. THe authors are strongly recommended to do such work.
Round 2
Reviewer 2 Report
Revisions are still needed to address the following points.
- Please provide detailed information about heat-treating process in Section 2.1. The information should include heat-treating temperature, time and cooling method, not limited to heating rate.
- The authors changed the scale bars in Figure 2 and Figure 5, which brought up a question about consistency of length scales in the figures. It seems that Figure 2(f) was taken at a different magnification as the length scale of dendrite arms should not vary greatly during heat treatment.
- The upper figures in Figure 5(b) and (c): “FCC with BCC needles” should be pointed to specific features.
- Figure 6 caption: Which citation is it referred to?
- Colorbars are needed for Figure 7(b) and (c).
- Figure 9: It is not rational to connect the points that represent concentration of individual elements. A bar chart should be used. The legends for both plots are misleading.
- A careful language check is needed.
Reviewer 3 Report
The paper was significantly improved and it can be published.
Author Response
Thank you for reviewing the manuscript.